# Fucosyltransferase 2 inhibitors: Identification *via* docking and STD-NMR studies

**Humaira Zafar**[1]*, **Muhammad Atif**[2], **Atia-tul-Wahab**[1], **M. Iqbal Choudhary**[1,2,3]

**1** Dr. Panjwani Center for Molecular Medicine and Drug Research, International Center for Chemical and Biological Sciences, University of Karachi, Karachi, Pakistan, **2** H. E. J. Research Institute of Chemistry, International Center for Chemical and Biological Sciences, University of Karachi, Karachi, Pakistan, **3** Faculty of Science and Technology, Department of Chemistry, Universitas Airlangga, Komplek Campus C, Surabaya, Indonesia

* hamramalik@gmail.com, humaira.zafar@iccs.edu

**Data Availability Statement:** All relevant data are within the paper and its Supporting Information files.

## Abstract

Fucosyltransferase 2 (FUT2) catalyzes the biosynthesis of A, B, and H antigens and other important glycans, such as (Sialyl Lewis$^x$) sLe$^x$, and (Sialyl Lewis$^y$) sLe$^y$. The production of these glycans is increased in various cancers, hence to design and develop specific inhibitors of FUT2 is a therapeutic strategy. The current study was designed to identify the inhibitors for FUT2. *In silico* screening of 300 synthetic compounds was performed. Molecular docking studies highlighted the interactions of ligands with critical amino acid residues, present in the active site of FUT2. The epitope mapping in ligands was performed using the STD-NMR experiments to identify the interactions between ligands, and receptor protein. Finally, we have identified 5 lead compounds **4**, **5**, **26**, **27**, and **28** that can be studied for further development as cancer therapeutic agents.

## 1. Introduction

Fucosyltransferases are the bi-substrate enzymes, belonging to the glycosyltransferase family. They catalyze the transfer of L-fucose moiety to different sugar acceptors [1]. There are 11 fucosyltransferases, and all of them have the same donor substrate *i.e.*, GDP fucose, while differing in the position of fucose transfer, and the type of sugar acceptors [2,3]. In humans, α-1, 2 FUTs are specified as FUT1 (H-type enzyme), and FUT2 (Se-type enzyme) [4]. The FUT2 is located on the epithelial cells as well as in body fluid [5]. It is involved in the synthesis of blood group antigens (A, B, and H), by catalyzing the transfer of fucose moiety to the terminal galactose of *N*-acteylactosamine (NLC) in α-1, 2 linkages [6–8]. The blood group antigens and precursors or related antigens such as Lewis y (Le$^y$), sialyl Lewis a (sLe$^a$), sialyl Lewis x (sLe$^x$), and Lewis x [9] have been reported to correlate with breast tumor progression. Increased levels of FUT2 promote tumor growth in the breast [10], bladder, ovaries, lungs, and prostate [11], and consequently stimulates uncontrolled cellular proliferation, adhesion, invasion, and metastasis [12].

The development of FUTs inhibitors is an appropriate strategy to treat various types of cancers. Despite the rigorous efforts for the development of FUT inhibitors, only a limited

**Funding:** Dr. Panjwani Center for Molecular Medicine and Drug Research, 2310-2018, Dr. Humaira Zafar.

**Competing interests:** The authors have declared no competing interest exist.

**Abbreviations:** FUT2, Fucosyltransferase 2; STD-NMR, Saturation Transfer Difference NMR Studies; NEM, *N*-Ethylmaleimide; GDP-Fuc, Guanosine Diphosphate Fucose; MM-GBSA, Molecular Mechanics Generalized Born Surface Area; PDB, Protein Data Bank; FBDD, Fragment-based drug discovery; MD, Molecular Dynamic Simulation; BLAST, Basic Local Alignment Search tool; FUTs, Fucosyltransferases; sLe^a, Sialyl Lewis a; sLe^x, Sialyl Lewis x; NLC, *N*-Acteyllactosamine; GDP, Guanosine Diphosphate.

outcome has been reported [13]. There are many reasons for this marginal success; the key reasons include the lack of some FUTs crystal structures, complex transition state of FUTs reaction, and low binding affinity for acceptor ligands [14].

Fragment-based drug discovery (FBDD) is an important approach, progressively applied in hit identification [15–17]. In particular, FBDD identify low molecular weight compounds that serve as starting points for drug designing against biomolecular drug targets [18,19]. Several drugs have been developed through the FBDD approach [20], and approved by US-FDA [21]. We have performed the computational fragment-based approaches, such as molecular docking studies, followed by the biophysical techniques *i.e.*, STD-NMR experiments for the evaluation of fragment hits for FUT2 inhibition.

## 2. Experimental Section

### 2.1 Chemicals

Guanosine 5′-diphosphate-β-L-fucose sodium salt (Cat. No. G4401), *N*-acetyl-D-lactosamine (Cat. No. A7791), DMSO HPLC grade (Cat. No. 34869–2.5L), *Tris* (hydroxymethyl) aminomethane (Cat. No. 106B), hydrochloric acid (Cat. No. H1758), *N*-ethylmaleimide (Cat. No. E3876), Deuterium oxide (Cat. No. 014100.2050), deuterated Tris (Cat. No. DLM- 1814–5), and deuterated DMSO (Cat. No. 015100.2040) were purchased from Deutero GmbH, Germany. Recombinant fucosyltransferase 2 (FUT2) (Cat. No. RPF192Hu01) was purchased from Cloud-Clone Corp. (CCC, USA).

### 2.2 Homology modeling

Homology modeling was performed individually for the donor and acceptor sites of FUT2 [22]. This was due to a low sequence similarity between the human FUT2 with other proteins deposited in the Protein Data Bank (PDB). Sequence alignment was carried out using the BLAST tool, and acceptor and donor binding sites for FUT2 were designed using *in-silico* studies [23,24].

### 2.3 Ligand preparation

The ligand structures were taken from the Molecular Bank of Dr. Panjwani Center for Molecular Medicine and Drug Research (*PCMD*). Ligands were prepared using the *LigPrep* tool (Schrödinger) by modifying the torsions of the ligands, and assigning them appropriate protonation states [25,26].

### 2.4 Molecular docking studies

Molecular docking studies were performed using the Glide 6.9 module of the Schrödinger suite of programs [27,28]. The crystal structures of *in-silico* mutated proteins (1W3F and 3ZY5) were used for the ligand docking studies [29]. The site for docking was defined using a grid box of dimensions 10x10x10 Å around the centroid of the co-crystallized ligand *N*-acetyl-lactoseamine, (NLC) in 1W3F, while GDP-fucose in 3ZY5 [30]. Ligands were docked using the Glide XP module, and the best-docked poses were used for the interpretation of final results [31,32].

### 2.5 MM-GBSA Tool for binding energy estimation

Docking is a tool used to identify the best orientation of the molecules (protein-ligand) to bind, and form the stable complex. To rank the docked poses, and explore the binding affinity, prime MM-GBSA Schrödinger tool 2019–2 was used [33]. MM-GBSA is generally used to

approximate the binding affinity of the ligands [34]. This tool was performed to re-rank the docked conformations of each listed fragment, received by the Glide XP dock tool, and to approximate the relative binding affinity of fragments (ligands). A more negative value of binding energy (presenting in kcal/mol) specified stronger binding affinity [35,36].

## 2.6 ADME Properties and toxicity studies of fragments

SwissADME online analysis (Swiss Institute of Bioinformatics, Switzerland), was carried out for the fragments that were selected as potential drug candidates [37]. The toxicity of these fragments was predicted by the online tool for the toxicity prediction of small molecules, *i.e.*, ToxiM using the descriptors prediction model of the MetaBiosys [38,39].

## 2.7 STD-NMR Screening experiments

All the STD-NMR experiments were performed on Bruker 600 MHz NMR instrument at 298 K using Stddiffesgp.3 pulse program Saturation time was 3 s, while interpulse delay (D1) was the same as D20 or D20 + 1 [40]. The loop counter was 8.0 and 4.0. STD-NMR spectra were recorded with 1024 scans (NS). For each experiment, a 90° pulse was calibrated separately. Gaussian selective pulses of 48 ms length with an excitation bandwidth of 140 Hz, separated by 1 ms delays were used [39,41]. To saturate the protein selectively, on-resonance irradiation was provided at -2.2 ppm (protein resonances), while off-resonance irradiation was provided at 30 ppm. The difference spectrum was obtained by subtracting the on-resonance irradiation spectrum from the off-resonance spectrum [42].

Quantification of STD effects was carried out by using the STD amplification factor ($A_{STD}$) for a given saturation time (3.0 s) at the given substrate concentration. $A_{STD}$ is defined as:

$$(I_0 - I_{STD})/I_0 * \text{ligand excess.}$$

STD Spectra were obtained by subtracting the on-resonance from the corresponding off-resonance spectrum. STD effects were calculated using $(I_0—I_{STD})/I_0$, in which the term ($I_{STD}$) defines the peak intensity in the STD spectrum, and $I_0$ the peak intensity in the off-resonance spectrum. The resulting most intensive STD effect in each spectrum was allocated to 100%. The remaining STD signals were referenced to this most intensive signal.

## 3. Results and discussion

FUT2 is involved in a variety of physiological and pathological processes, such as cell adhesion, cell signaling, and breast tumor progression [2,43]. Overexpression of FUT2 is associated with inflammation and cancers. To date, there is no marketed drug available for FUT2 inhibition. During the current study, we screened various libraries of the fragment like molecules using molecular docking studies. These fragment libraries were obtained from the *in-house* Molecular Bank of Dr. Panjwani Center for Molecular Medicine and Drug Research (*PCMD*).

## 3.1 Design of *in-silico* homology model of FUT2

Crystal structure of POFUT1 in complex with GDP-fucose (3ZY5), and hemolytic lectin from the mushroom, *Laetiporus sulphreus* with *N*-acteyl-lactoseamine *NLC* (1W3F were selected for donor and acceptor sites, respectively. On the basis of sequence alignment results, the amino acid residues of the active site (1W3F, and 3ZY5) were replaced with residues of FUT2. The ligands (GDP-fucose and *NLC*) were re-docked, and interactions were compared with that of the native proteins (**Figs 1 and 2**). Most of the interactions were retained in mutated donor and acceptor sites, and the results were further validated by molecular dynamic simulation studies (**S1 Fig**).

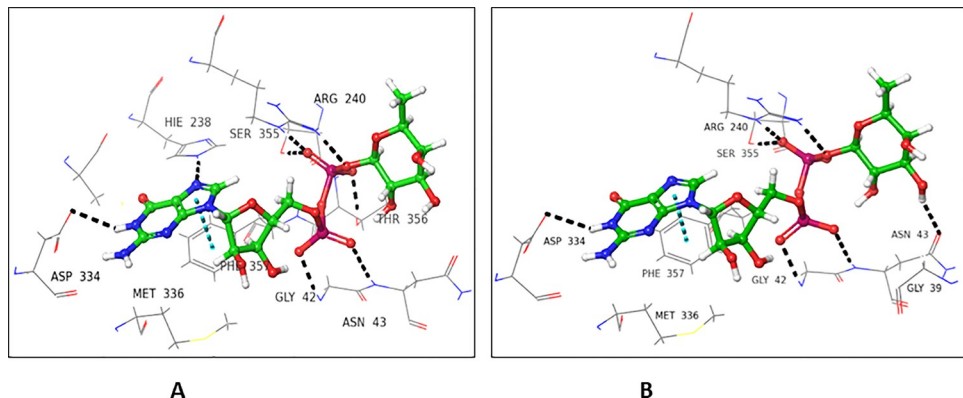

**Fig 1.** GDP fucose interactions with **(A)** native, and **(B)** mutated 3ZY5.

## 4. Molecular docking studies

### 4.1 Molecular docking studies of standard inhibitor *N*-ethylmaleimide

*N*-Ethylmaleimide (NEM) is the standard inhibitor for several human FUTs. This is a cysteine-specific modifying reagent that affects FUT2 activity. Till date there is no clinical drug available for FUT2 inhibition, therefore, NEM was used as a standard lab inhibitor in this study [44].

### 4.1.1 Interactions of *N*-ethylmaleimide with FUT2

*N*-Ethylmaleimide (NEM) showed a higher docking score for the donor substrate (-4.317), while the acceptor substrate showed a comparatively lower docking score (-2.395) for FUT2. In the donor binding domain, Arg240 and Ser355 interacted with carbonyl oxygen at C-2 *via* H-bond, while Asn43 form H-bond with carbonyl oxygen at C-5. These interactions are catalytically important, as Ser355 and Arg240 are involved in the binding of GDP-fucose (**Fig 3**).

In case of the acceptor binding domain, Trp126 and Arg123 form H-bond with carbonyl oxygen located at C-2. Arg76 interacted with carbonyl oxygen *via* H-bond at C-5. Arg123 and

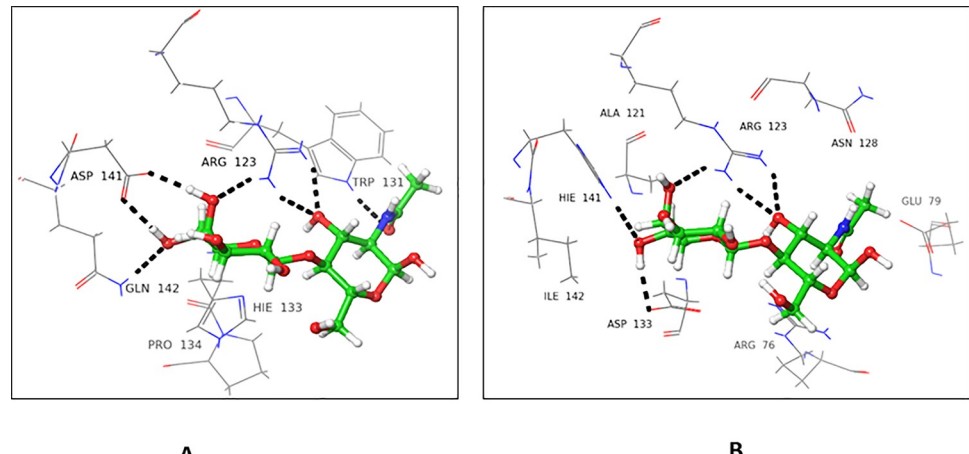

**Fig 2.** NLC Interactions with **(A)** native, and **(B)** mutated 1W3F.

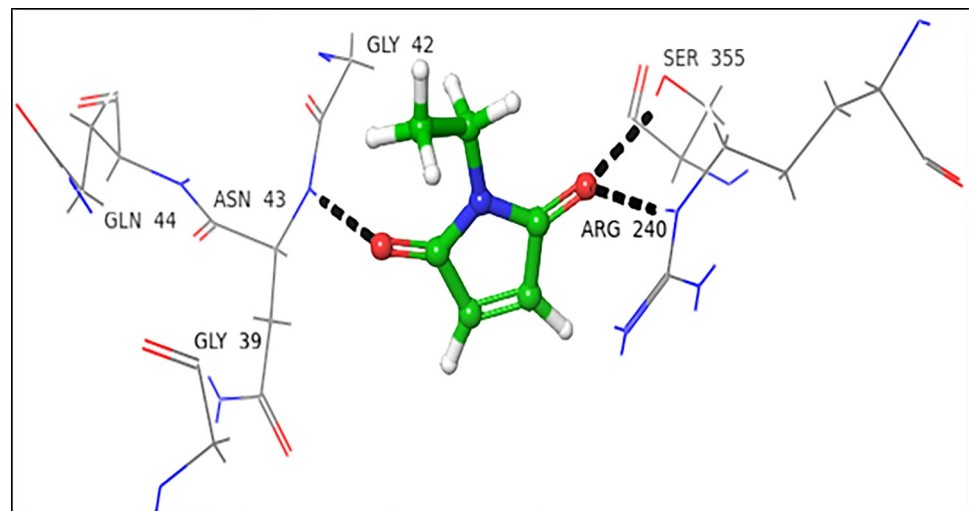

**Fig 3. Interactions of NEM with donor active site 3ZY5.** 3D Ligand interactions diagram: Black dashed lines represent hydrogen bonding.

Arg76 are among the catalytically important amino acid residues of FUT2, as they are involved in the binding of NLC (**Fig 4**).

**4.1.2 *In-silico* screening of synthetic libraries.** Over 300 synthetic fragment like molecules were screened *in-silico*, individually for acceptor and donor binding domains of FUT2,

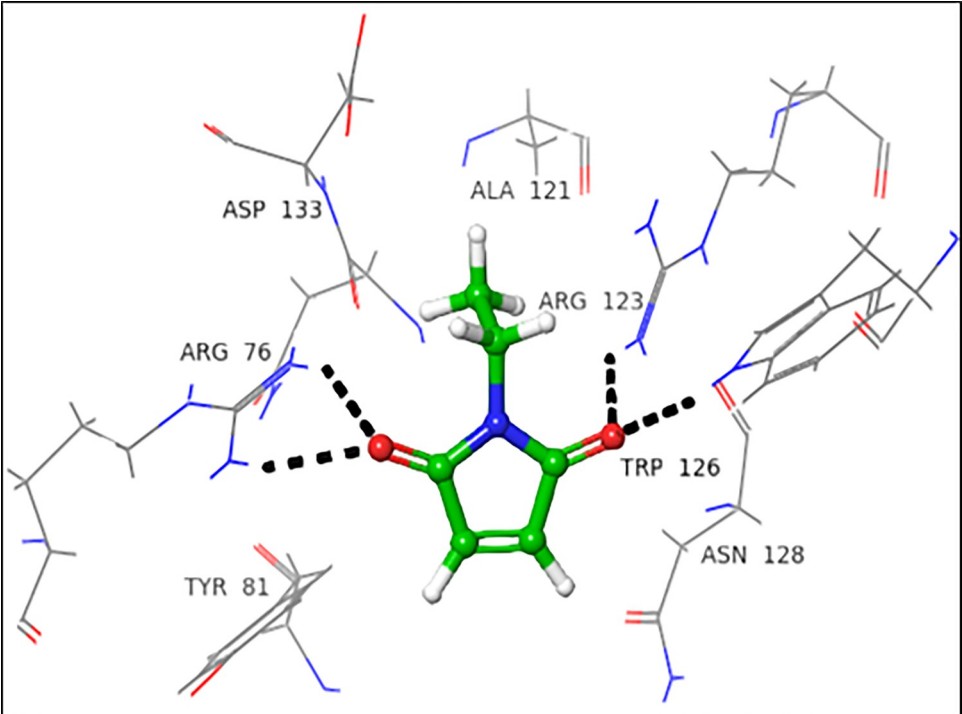

**Fig 4. Interactions of NEM with acceptor active site 3ZY5.** 3D Ligand interactions diagram; black dashed lines represent hydrogen bonding.

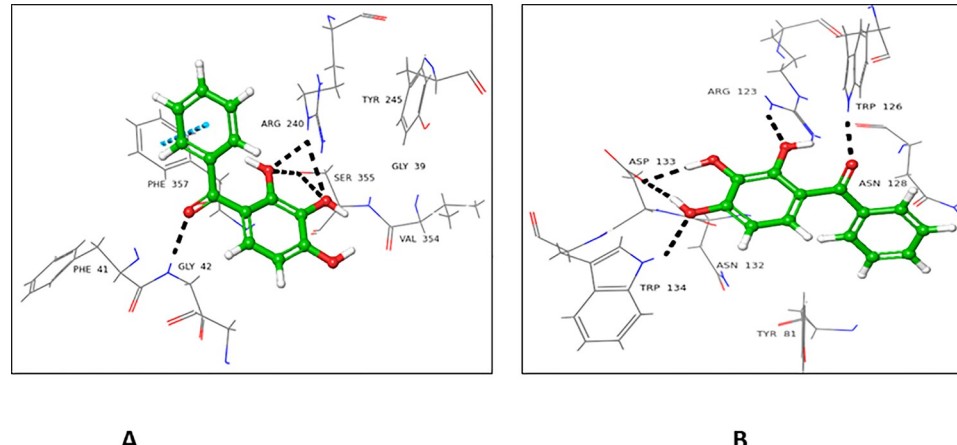

**A** **B**

**Fig 5.** Interactions of compound **1** with (**A**) donor and (**B**) acceptor active site. 3D Ligand Interactions diagram; blue color dashed lines showing π-π stacking bond, while black represents hydrogen bonds.

and docking scores were in the range of -9.08 to -2.30. Top-ranked 50 fragments are shown in **S1 Table** and their interactions with FUT2 are shown in **S2 Table**.

## 4.2. Ligand receptor interactions of selected fragments

**4.2.1 Phenyl-(2', 3', 4'-trihydroxyphenyl) methanone (1).** The docking result of ligand **1** showed an excellent network of interactions with donor and accepter domains of FUT2. Compound **1** showed a higher docking score for the donor binding domain (-6.983), in comparison to the acceptor binding domain (-3.541). Hydroxyl group attached at C-2' form H-bond with Arg240 and Ser355. Carbonyl oxygen located at C-1 interacted with Gly42 *via* H-bond, while π-π stacking interaction of phenolic ring to Phe357 of the donor active site of FUT2 was also observed (**Fig 5A**). In the acceptor binding domain, Asp133 established two H-bond with hydroxyl groups, attached at C-3 and C-4. Trp134 interacted with the hydroxyl group at C-4, while Arg123 form a H- bond with the hydroxyl group at C-2 (**Fig 5B**).

**4.2.2 N-(4-Amino-5-ethoxy-2-(methoxymethyl) phenyl) benzamide (2).** Compound **2** showed a higher docking score for the donor binding domain (-8.217), in comparison to the acceptor binding domain (-7.412). Carbonyl oxygen at C-1 interacted with His238 residue of FUT2 in the donor binding site *via* H-bond (**Fig 6A**). In case of the acceptor binding domain, H-bond and π-cation bonding interactions were observed. Arg123 interacted with carbonyl oxygen *via* H-bond at C-1, as well as established π-cationic interactions with compound **2** (**Fig 6B**).

**4.2.3 5-(Benzyloxy)-1H-indole-2-carboxylic acid (3).** Compound **3** showed a higher docking score for the donor active site (-7.462), while a lower docking score for the acceptor active site (-6.167). In the donor site, Thr356 interacted with carboxyl oxygen atom *via* H-bond at C-2. Phe357 formed π-π stacking interaction with compound **3** (**Fig 7A**). In case of the acceptor binding site, Arg76 and Trp134 form H-bond with carboxyl group at C-2. Asp133 interacted *via* H-bond, while Arg123 form π-cationic interaction with indole ring of compound **3** (**Fig 7B**).

**4.2.4 3-Hydroxy-2-methyl-4H-pyran-4-one (4).** Compound **4** showed almost similar docking scores for the donor (-3.556) and acceptor (-3.256) binding domains of FUT2. In the donor site, Ser355 and Arg240 interacted through H-bonding with carbonyl oxygen at C-4, while oxygen at C-1 form H-bond with Asn43 (**Fig 8A**). In the acceptor site, Trp126 and

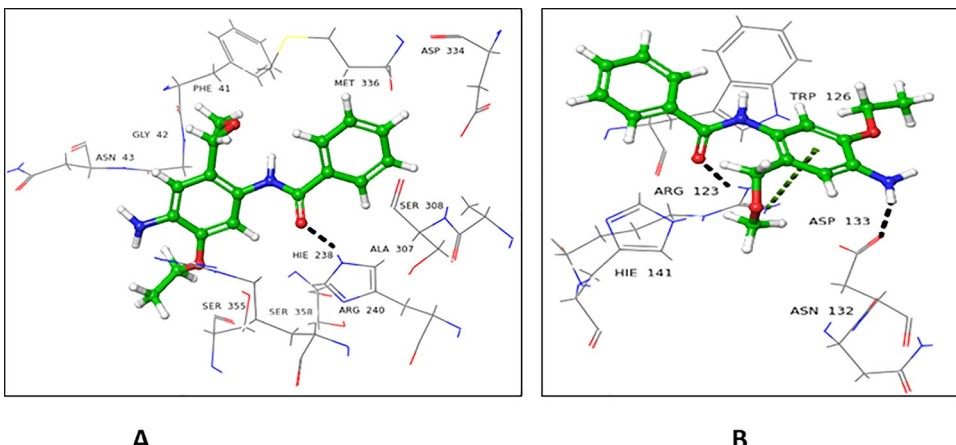

**Fig 6.** Interactions of compound **2** with (**A**) donor and (**B**) acceptor site. 3D Ligand interactions diagram; black dashed lines show hydrogen bonds; green dashed lines indicate π-cation interactions.

Arg123 interacted through H-bonding with carbonyl oxygen at C-1. Asn128 established H-bond with a hydroxyl group at C-3 (**Fig 8B**).

**4.2.5 2-Hydroxynaphthalene-1, 4-dione (5).** Compound **5** showed a lower docking score for the donor binding domain (-3.256), in comparison to the acceptor binding domain (-5.078). Arg40 and Asn43 established H-bonds with the carbonyl oxygen at C-1. Gly42 interacted with the carbonyl oxygen through H-bond at C-2. Carbonyl oxygen at C-4 bind with Arg240 and Ser355 through H-bonds. Phe261 linked *via* π-π stacking interaction with the phenyl ring of compound **5** (**Fig 9A**). All these amino acid residues were involved in the binding of GDP-fucose, and therefore play a critical role in the enzymatic reaction of FUT2. Compound **5** interacted with acceptor site amino acid residues, mainly *via* H-bond. The oxygen of carbonyl at C-1 form H- bond with Arg123. Hydroxyl group at C-2 interacted with Asp133 and His141*via* H-bonding (**Fig 9B**).

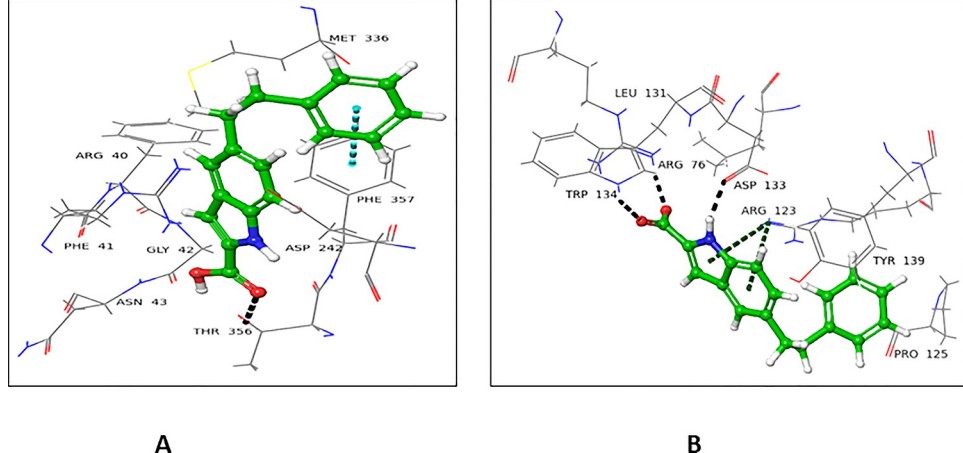

**Fig 7.** Interactions of compound **3** with (**A**) donor and (**B**) acceptor site. 3D Ligand interactions diagram; black dashed lines indicate hydrogen bonds; green dashed lines indicate π-cation interactions, while blue show π-π stacking interactions.

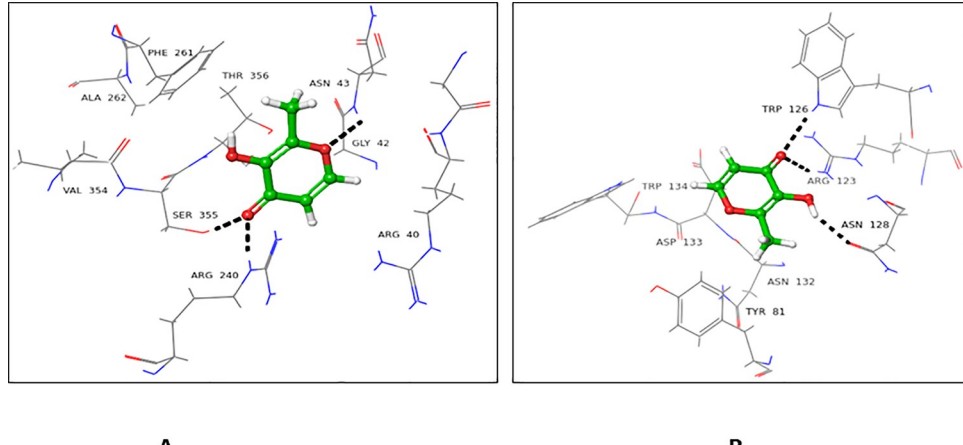

**Fig 8.** Interactions of compound **4** with (**A**) donor and (**B**) acceptor active site. 3D Ligand interactions diagram; black dashed lines represent hydrogen bonds.

**4.2.6 2-(Carboxymethyl)-benzoic acid (26).** Compound **26** showed a higher docking score for the donor active site (-5.132), as compared to the acceptor active site (-3.476). C-1' interacted with Asn43 *via* H-bond, while phenyl ring showed π-cation interaction with Arg240 (**Fig 10A**). In the case of the acceptor active site. Arg76 form H-bond with OH at C-1', Asp133 established H-bond with carboxyl group at C-1' and C-7. Arg123 form H-bond at C-2' as well as π-cation interaction with the benzoic ring (**Fig 10B**).

**4.2.7 3-Hydroxy-4-nitrobenzoic acid (27).** Compound **27** showed a higher docking score for the donor active site (-5.546), in comparison to the acceptor active site (-3.097). Asp334 interacted with the hydroxyl groups at C-7 of ligand *via* H-bond. The nitro group at C-4 form H-bond with His238 (**Fig 11A**). In case of the acceptor binding domain of FUT2, Asp133 and His141 form H-bonding with hydroxyl groups at C-7 and C-3, Arg123 interacted with nitro group at C-4 *via* H-bond, while Tyr139 and Trp126 interacted with $NO_2$ at C-4, and Arg123 showed π-cationic interactions with the phenolic ring of compound **27** (**Fig 11B**).

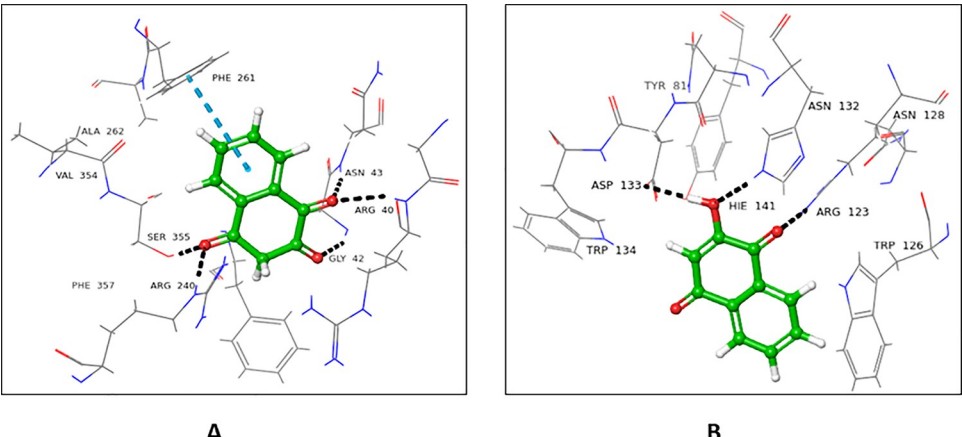

**Fig 9.** Interactions of compound **5** with the (**A**) donor and (**B**) acceptor domain of FUT2. 3D Ligand interactions diagram; blue dashed lines indicate π-π stacking bond, while black dashed lines show hydrogen bonding.

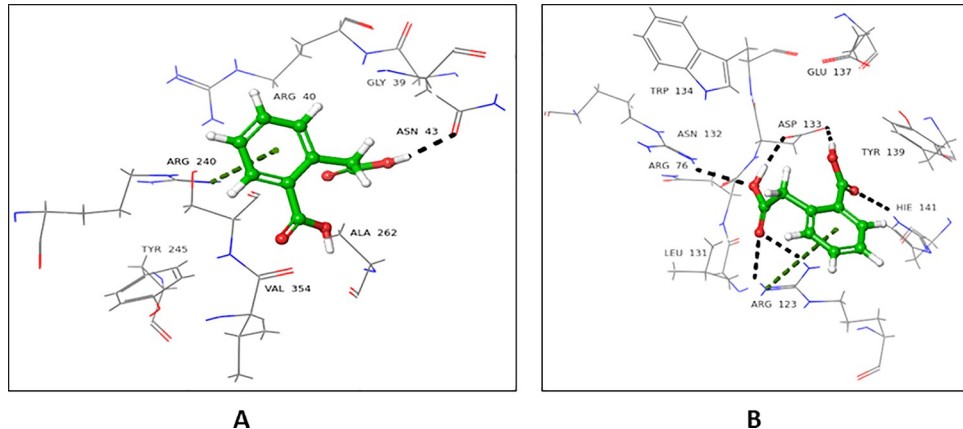

**Fig 10.** Interactions of compound **26** with the **(A)** donor and **(B)** acceptor domain FUT2. 3D Ligand interactions diagram; green dashed lines indicate π-cation bond, while black dashed lines represent hydrogen.

**4.2.8 4-(4'-Hydroxyphenyl) butan-2-one (28).** Compound **28** showed a higher docking score for donor active site (-5.123), while a lower docking score for acceptor active site (-2.553). Gly42 interacted with carbonyl oxygen at C-2 *via* H-bond. Asp334 form H-bond with a hydroxyl group at C-4', while Phe357 showed π-π stacking interactions with phenyl ring (**Fig 12A**). In case of acceptor active site, Arg123 interacted with carbonyl oxygen *via* H-bond at C-2. Asp133 forms H-bond with a hydroxyl group at C-4', while Trp126 established π-π stacking interaction with compound **28** (**Fig 12B**).

**4.2.9 (S)-2-Amino-3-(3,4-dihydroxyphenyl) propanoic acid (29).** Compound **29** showed a higher docking score for the donor active site (- 6.12), and a comparatively lower docking score for the acceptor active site (-3.76). Compound **29** comprises both aromatic as well as aliphatic protons. Aromatic and aliphatic protons of ligand **29** have no H -bonding, π -π stacking, or π-π cation interaction for the donor and acceptor active site. Consequently, compound **29** was found to be non-binder for the receptor protein (**Fig 13A and 13B**).

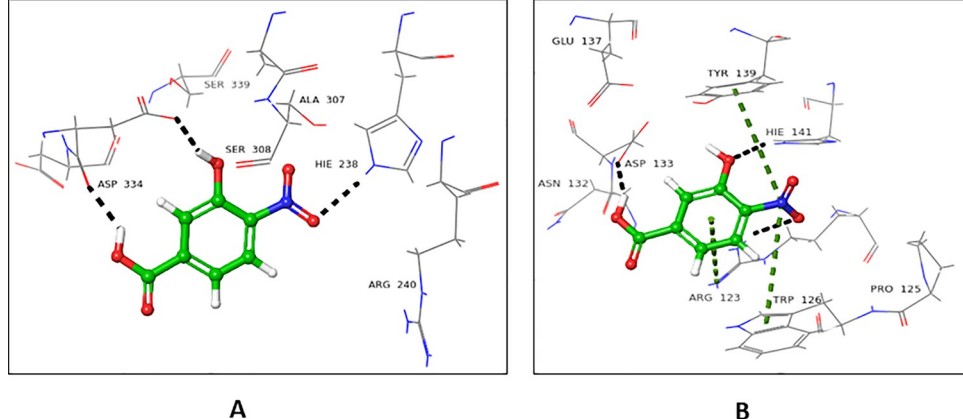

**Fig 11.** Interactions of compound **27** with the **(A)** donor and **(B)** acceptor domain of FUT2. 3D Ligand interactions diagram; black dashed lines representing hydrogen bonding bond.

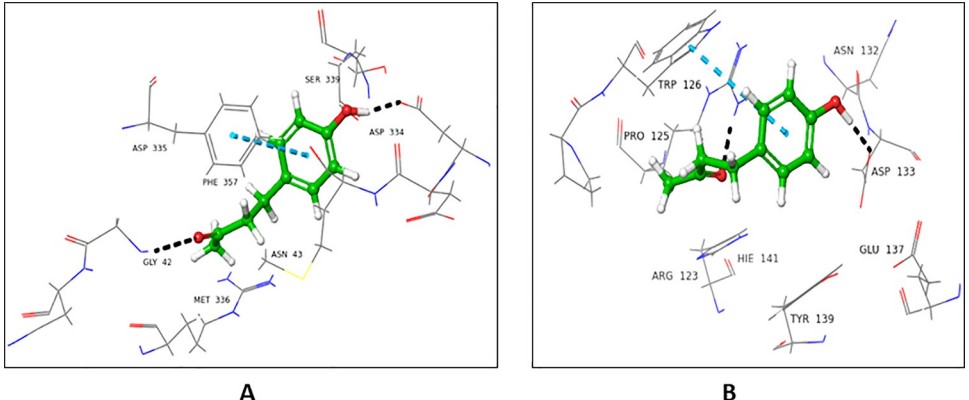

**Fig 12.** Interactions of compound **28** with the **(A)** donor and **(B)** acceptor domain of FUT2. 3D Ligand interactions diagram; blue dashed lines indicate π-π stacking bond while black dashed lines show hydrogen bond.

## 4.3 Saturation Transfer Difference NMR studies (STD-NMR)

Molecular docking studies were performed as a primary tool for the identification of new inhibitors of a clinically important enzyme, FUT2. Compounds with significant docking scores and binding interactions were selected for the study of ligand-receptor interactions at the atomic level *via* epitope mapping through STD-NMR spectroscopy. Individual ¹H-NMR spectrum of all the ligands **1–5**, **27–29**, and GDP-fucose in deuterated methanol and deuterated buffer are provided in supplementary file (**S2–S12** Figs).

**4.3.1 Donor activesite; GDP- fucose interactions with FUT2.** The epitope mapping of the donor substrate, GDP-fucose was performed. The STD-NMR spectrum showed that the fucose moiety of the donor substrate was significantly involved in interaction with the receptor protein. H-6" showed 100% saturation transfer, indicating that they were in close contact to the receptor protein displaying highest intensity signal, as compared to the remaining protons (**Fig 14**). H-4" and H-5" exhibited 84%, H-2" received 72% while H-1" and H-3" showed 71.81% relative saturation transfer from the receptor protein. The results indicated that fucose moiety in GDP-fucose has a higher affinity, as it is in close proximity to the protein. While the ribose and guanine moieties did not show STD-effects indicating that they are far from the receptor protein (FUT2).

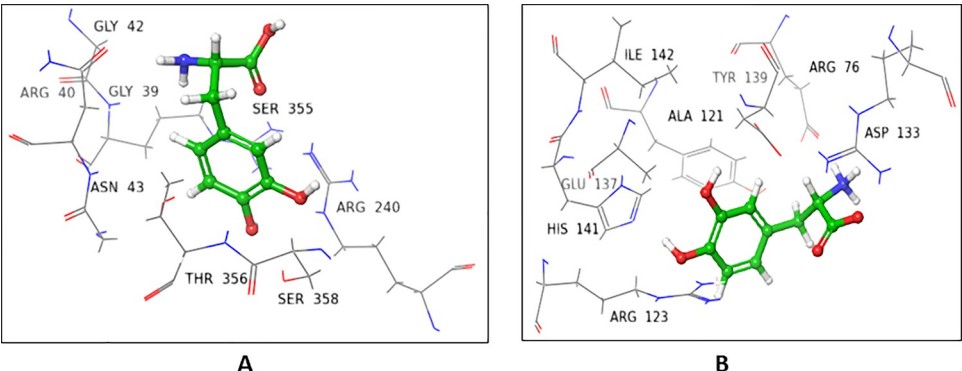

**Fig 13.** Interactions of compound **29** with the **(A)** donor and **(B)** acceptor domain of FUT2. 3D Ligand interactions diagram; Compound **29** for both Domains (donor, acceptor) have no interaction.

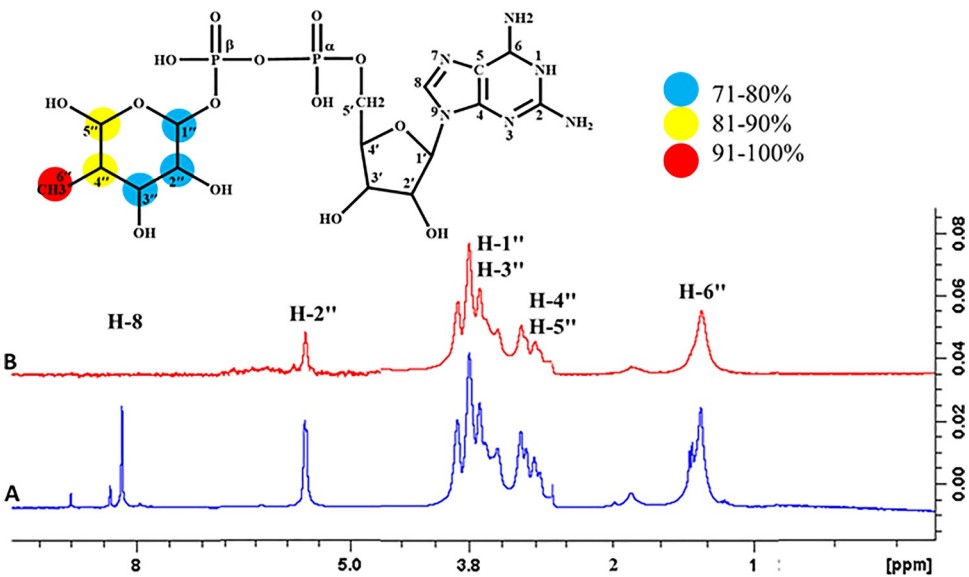

**Fig 14.** (**A**) Reference ¹H-NMR spectrum of GDP- Fucose. (**B**) Difference spectrum of GDP- Fucose with the enzyme.

**4.3.2 Standard inhibitor; *N*-ethylmaleimide (NEM) interactions with FUT2.** The epitope mapping was performed for the *N*-ethylmaleimide, which showed that all the protons of the compound were interacting with the receptor protein. For instance, H-3 and H-4 exhibited 100% saturation transfer indicating that they received maximum saturation from receptor proteins. Likewise, H-6 showed 64.81%, while H-7 showed 42.85% relative saturation transfer from the receptor protein (**Fig 15**).

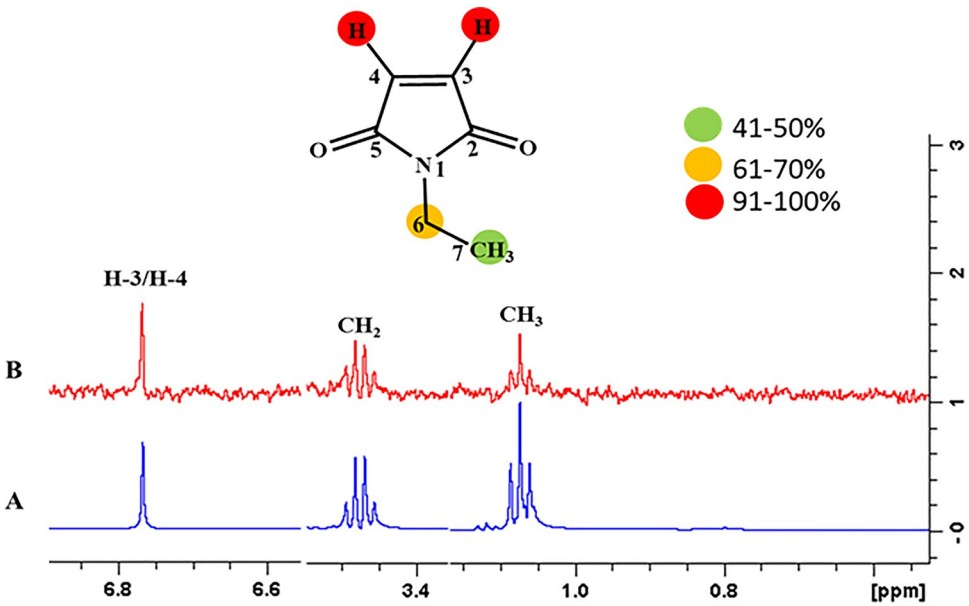

**Fig 15.** (**A**) Reference ¹H-NMR spectrum of standard inhibitor *N*-Ethylmaleimide. (**B**) Difference spectrum of *N*-Ethylmaleimide with the enzyme.

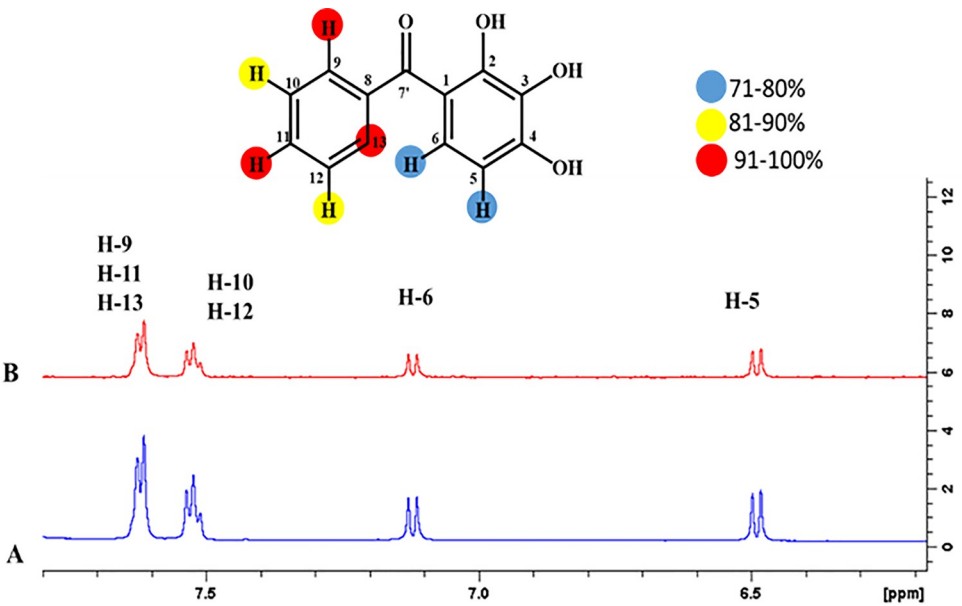

**Fig 16.** (**A**) Reference ${}^1$H-NMR spectrum of ligand **1**. (**B**) Difference spectrum of ligand **1** with the enzyme.

**4.3.3 Phenyl-(2, 3, 4-trihydroxyphenyl)-methanone (1).** The epitope mapping of ligand **1** showed that most of the protons were involved in binding with the receptor protein. For instance, H-9, H-11, and H-13 of ligand **1** exhibited 100% saturation transfer, indicating their proximity with the receptor protein. Similarly, H-10 and H-12 experienced 90% H-5 and H-6 showed 62% and 68% relative saturation transfer, respectively. (**Fig 16**).

**4.3.4 N-(4-Amino-5-ethoxy-2-(methoxymethyl)-phenyl)-benzamide (2).** Epitope mapping of ligand **2** displayed multiple proton interactions with the receptor protein. For instance, H-6 and CH$_3$ showed 100% saturation transfer, thus indicating their close proximity to the receptor proteins. H-3', H-4', and H-5' showed 91% H-2', and H-6' represented 86% H-3 showed 74%, while CH$_2$ showed 54% relative saturation transfer. Among all, the highest intensity signal, CH$_3$ showed its close proximity with the receptor protein (**Fig 17**).

**4.3.5 (S)-2-Amino-3-(3, 4-dihydroxyphenyl)-propanoic acid (3).** The epitope mapping of ligand **3** showed proton interaction with the receptor protein. H-3 showed 100% saturation transfer from the receptor protein. H-2' and H-6' showed 71%, H-3', H-5', and H-7 exhibited 74% relative saturation transfer. H-4 and H-4' showed 71% and 55% relative saturation transfer, respectively. Similarly, H-6 showed 67% -, while CH$_2$ showed 23% relative saturation transfer. Among them the highest intensity signals, H-3 showed its close contact with the receptor protein (**Fig 18**).

**4.3.6 3-Hydroxy-2-methyl-4H-pyran-4-one (4).** The epitope mapping of ligand **4** exhibited proton interactions with the targeted proteins. H-5 displayed 100% saturation transfer, indicating its close proximity with the receptor protein. H-6 showed 72%, while CH$_3$ received 62% relative saturation transfer (**Fig 19**).

**4.3.7 Hydroxynaphthalene-1, 4-dione (5).** The epitope mapping of ligand **5** showed proton interactions with the receptor protein. H-6 showed 100% saturation transfer Similarly, H-7 and H-5 exhibited 82and 87% relative saturation transfer, respectively, H-8 showed 48% relative saturation transfer. Among all the highest intensity signals, H-6 showed a close contact with their receptor protein (**Fig 20**).

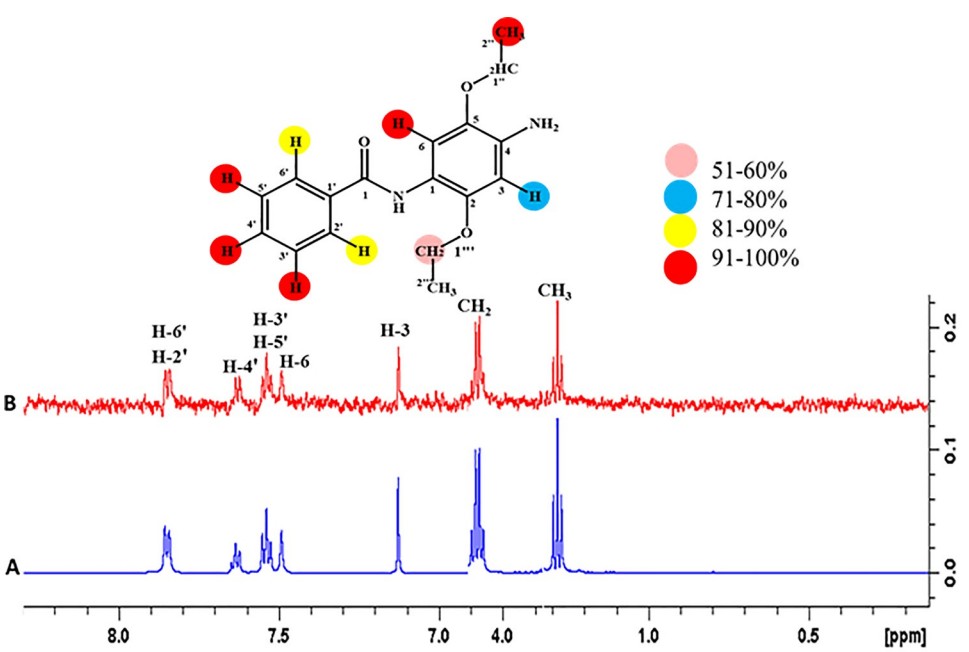

**Fig 17.** **(A)** Reference $^1$H-NMR spectrum of ligand **2**. **(B)** Difference spectrum of ligand **2** with enzyme.

**4.3.8 2-(Carboxymethyl)-benzoic (26).** Epitope mapping of ligand **26** showed the proton interactions with the receptor protein. H-3 of ligand displayed 100% saturation transfer indicating the maximum saturation from receptor protein. Similarly, H-3 and H-4 showed 71%, while H-5 received 60% relative saturation transfer. Among them, the highest intensity signal of H-3 showed close proximity with the receptor protein (**Fig 21**).

**4.3.9 3-Hydroxy-4-nitrobenzoic acid (27).** The result of epitope mapping showed interactions of ligand **27** protons with the receptor protein. H-2 showed 100% saturation transfer from receptor protein. Likewise, H-5 showed 42%, while H-6 exhibited 46% relative saturation transfer. The highest intensity signal of H-2 revealed its close contact with receptor protein (**Fig 22**).

**4.3.10 4-(4-Hydroxyphenyl)-butan-2-one (28).** Epitope mapping of ligand **28** displayed a network of interactions with the receptor protein. CH$_3$ showed 100% saturation transfer from the receptor protein. H-3' and H-5' showed 84%, while H-2', H-6', and H-4 displayed 77% relative saturation transfer, respectively. Similarly, CH$_2$ showed 69% relative saturation transfer. Among all the highest intensity signal, CH$_3$ showed its close proximity with the receptor protein (**Fig 23**).

**4.3.11 (S)-2-Amino-3-(3,4-dihydroxyphenyl) propanoic acid (29).** STD-NMR experiment was performed for compound **29** to analyze the binding affinity of the ligand with the receptor proteins. The STD spectra showed no peaks from the ligand indicating that this compound is a non-binder. The results were consistent with the docking studies as the docking results also showed no interaction of this compound with FUT2 (**Fig 24**).

## 5. Absorption, distribution, metabolism, excretion, and cytotoxicity studies of fragments

Effective and safe drugs exhibit a finely tuned combination of pharmacodynamics (PD) and pharmacokinetics (PK) properties, including high potency, affinity, and selectivity against the

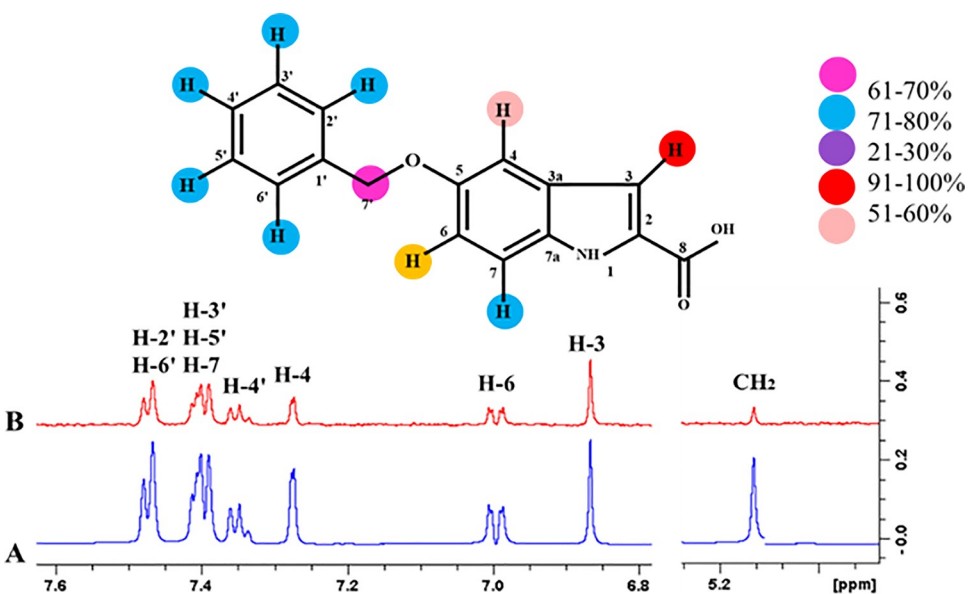

**Fig 18. (A)** Reference $^1$H-NMR spectrum of ligand **3**. **(B)** Difference spectrum of ligand **3** with the enzyme.

biomolecular target, along with adequate absorption, distribution, metabolism, excretion, and tolerable toxicity (ADME properties). The lipophilicity (Log*P*) and solubility (Log*S*) determine these properties. In our study, we identified that fragments **1–5**, and **26–28** all have a positive log*P* value. The positive value for log*P* denotes a higher concentration in the lipid phase (the compound is more lipophilic). The lipophilicity is helpful in the penetration across the vital membranes and biological barrier mechanism. All fragments showed the negative Log*S* value, indicating good solubility in the biphasic solvent condition. The solubility indicates how these drug candidates disseminate throughout the body. The result also suggested that all the

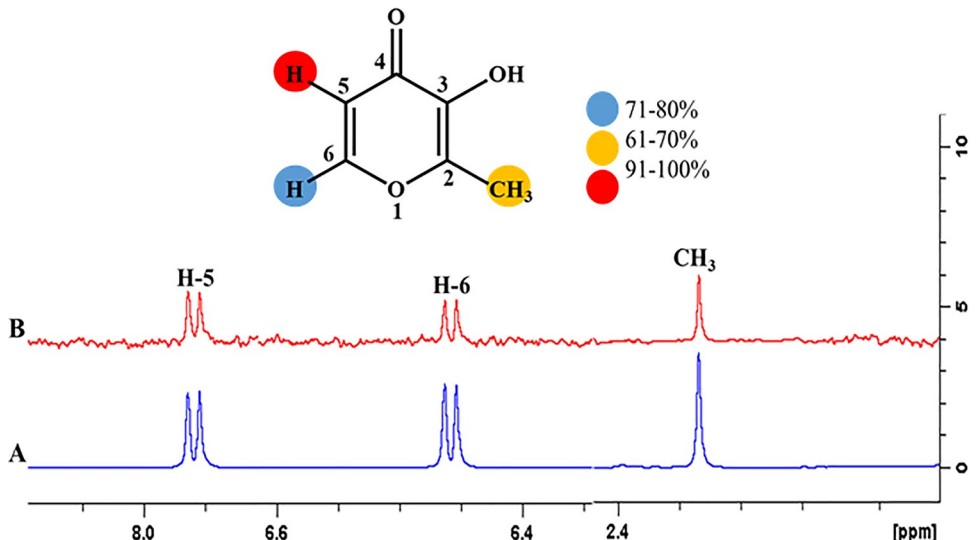

**Fig 19. (A)** Reference $^1$H-NMR spectrum of ligand **4**. **(B)** Difference spectrum of ligand **4** with the enzyme.

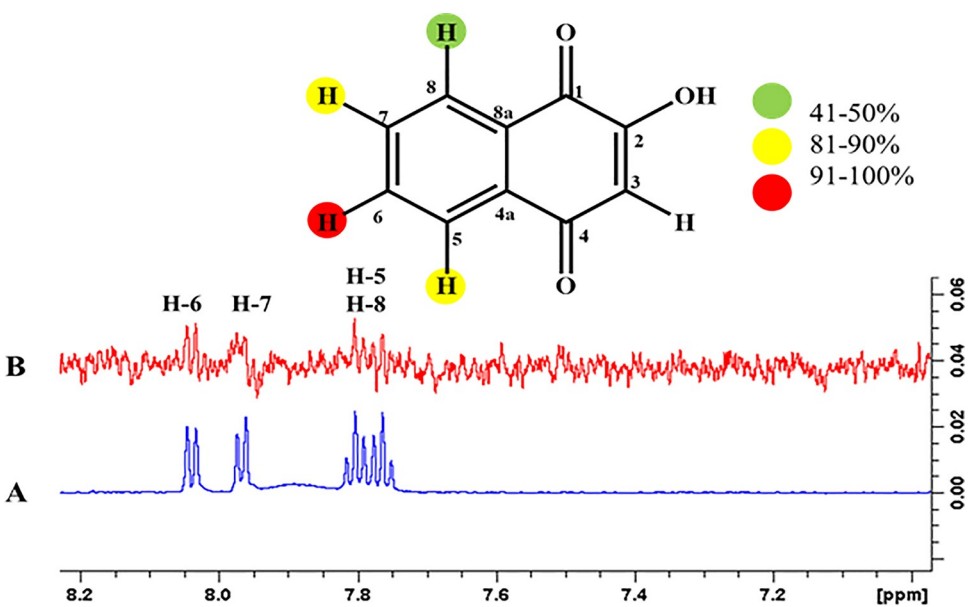

**Fig 20.** (A) Reference ¹H-NMR spectrum of ligand **5**. (B) Difference spectrum of ligand **5** with enzyme.

fragments **1–5**, **26–28** follow *Lipinski's* rule of five indicate that all these fragments have drug-likeness properties, and no violations were found.

Toxicity is one of the most important concerns with drugs. A value equal to 0.8 or more is considered as a toxic fragment. In our studies, fragments **1–3** showed a slightly high value of toxicity, while fragments **4**, **5**, **26**, **27**, and **28** showed no toxicity (**Table 1**).

## 6. Conclusion

Overexpression of fucosylated glycans, such as sLe^x and sLe^y, has been well correlated in various cancers, specifically in breast tumor progression. During the current study, *in-silico* and

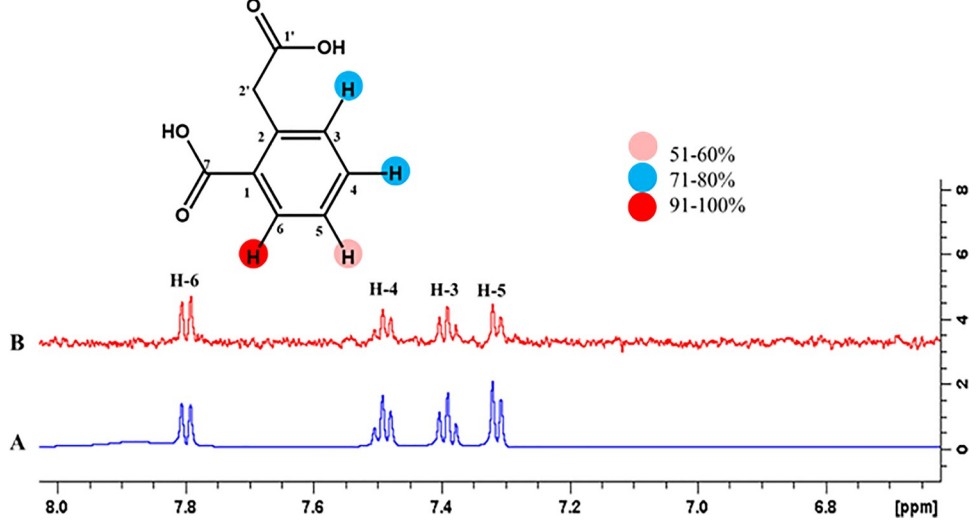

**Fig 21.** (A) Reference ¹H-NMR spectrum of ligand **26**. (B) Difference spectrum of ligand **26** with enzyme.

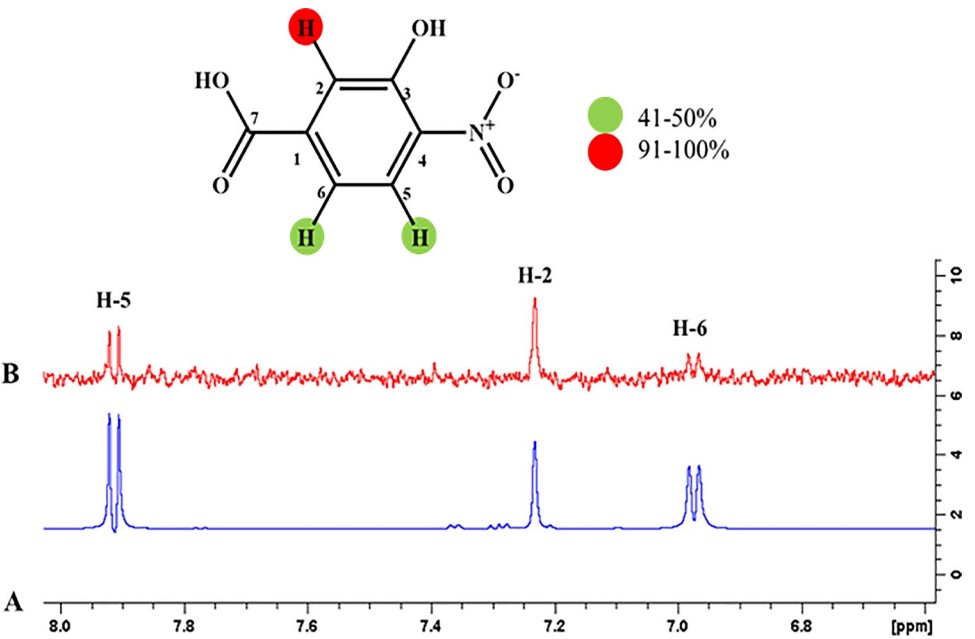

**Fig 22.** **(A)** Reference $^1$H-NMR spectrum of ligand **27**. **(B)** Difference spectrum of ligand **27** with enzyme.

STD-NMR studies were carried out to identify small molecular inhibitors of FUT2. Among 300 synthetic compounds (fragments), 50 fragments were selected for the ligand- receptor interactions using molecular docking studies. Further selection on the basis of binding energies identified 8 ligands **1**–**5**, **26**, **27**, **28,** and **29** that were selected for STD-NMR studies and

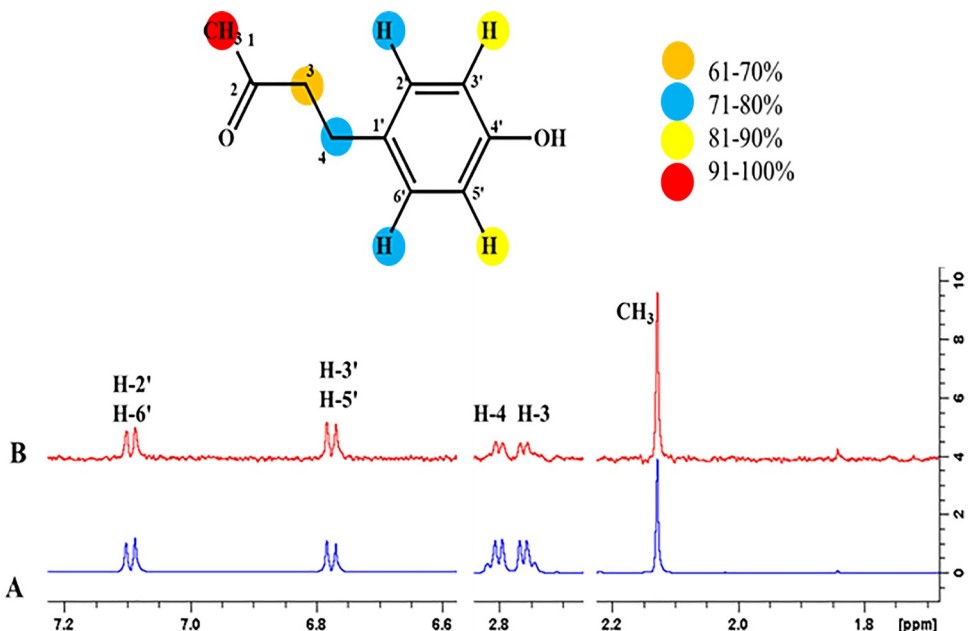

**Fig 23.** **(A)** Reference $^1$H-NMR spectrum of ligand **28**. **(B)** Difference spectrum of ligand **28** with the enzyme.

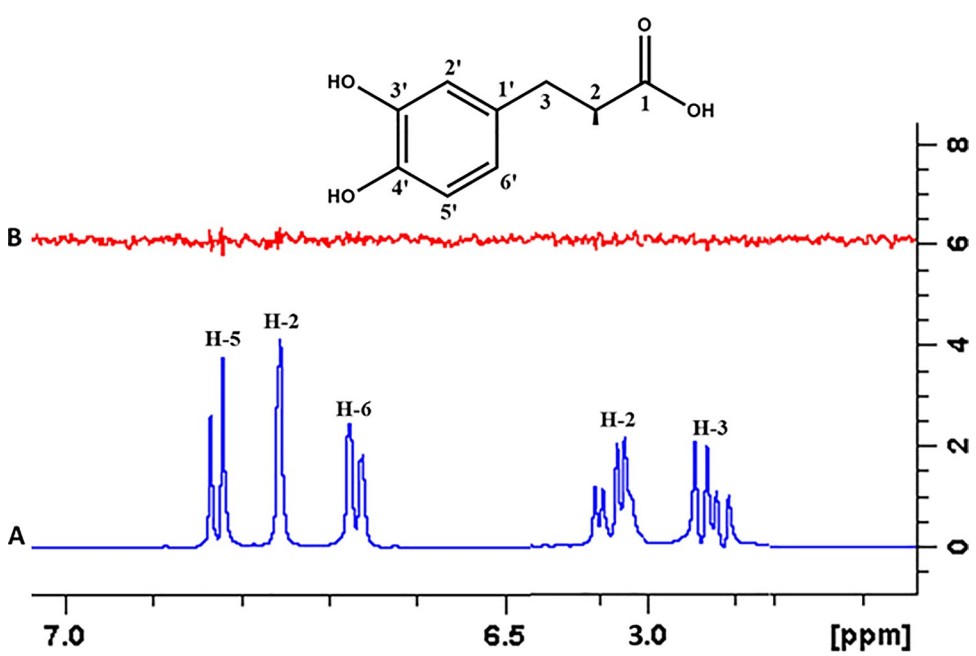

**Fig 24.** **(A)** Reference $^1$H-NMR spectrum of ligand **29**. **(B)** Difference spectrum of ligand **29** with the enzyme.

**Table 1.** **Fragments absorption, bioavailability, and toxicity prediction, followed by Lipinski's rule of five.**

| Fragments | Molecular weight | Docking score | | Toxicity | LogP | LogS | H-bond Donor | H-bond Acceptor | Druglikeness/Lipinski rule |
|---|---|---|---|---|---|---|---|---|---|
| | | 3ZY5 | 1W3F | | | | | | |
| Fragment **1** | 230.22 | -9.08 | -8.54 | 0.866 | 2.50 | -3.43 | 3 | 4 | Yes/no violation |
| Fragment **2** | 300.35 | -8.21 | -7.41 | 0.939 | 2.80 | -3.38 | 2 | 3 | Yes/no violation |
| Fragment **3** | 265.31 | -7.46 | -6.16 | 0.96 | 3.52 | -4.63 | 2 | 2 | Yes/no violation |
| Fragment **4** | 126.11 | -5.55 | -5.25 | 0.719 | 2.55 | -1.17 | 1 | 3 | Yes/no violation |
| Fragment **5** | 174.15 | -5.25 | -5.07 | 0.778 | 1.21 | -2.13 | 1 | 3 | Yes/no violation |
| Fragment **26** | 180.18 | -7.13 | -9.47 | 0.588 | 1.11 | -2.28 | 2 | 4 | Yes/no violation |
| Fragment **27** | 183.12 | -6.54 | -7.09 | 0.647 | 0.36 | -2.43 | 2 | 5 | Yes/no violation |
| Fragment **28** | 164.20 | -6.12 | -6.55 | 0.756 | 1.84 | -1.96 | 1 | 2 | Yes/no violation |

epitope mapping. Drug-like properties of these selected ligands, including absorption, distribution, metabolism, excretion, and toxicity of the fragments, *via* swissADME and ToxiM analysis, further supported them as potential drug candidates. This study has identified 5 potential lead molecules **4**, **5**, **26**, **27**, and **28** that can be further studied for *in-vitro* FUT2 inhibition and kinetic assays.

## Supporting information

**S1 Fig. Protein ligand RMSD: Simulation of the part of the A. donor (3ZY5) and B. acceptor (1W3F).** Red color indicates the ligand RMSD, while blue is for protein. During the total run time of simulation, the acceptor and donor were stable in their respective pockets. (DOCX)

**S2 Fig. $^1$H-NMR of receptor-ligand complex.** Resonances of ligand (GDP-Fucose) are presented.
(DOCX)

**S3 Fig. $^1$H-NMR of receptor-ligand complex.** Resonances of ligand (*N*-Ethylmaleimide) are presented.
(DOCX)

**S4 Fig. $^1$H-NMR of receptor-ligand complex.** Resonances of ligand (ligand **1**) are presented.
(DOCX)

**S5 Fig. $^1$H-NMR of receptor-ligand complex.** Resonances of ligand (ligand **2**) are presented.
(DOCX)

**S6 Fig. $^1$H-NMR of receptor-ligand complex.** Resonances of ligand (ligand **3**) are presented.
(DOCX)

**S7 Fig. $^1$H-NMR of receptor-ligand complex.** Resonances of ligand (ligand **4**) are presented.
(DOCX)

**S8 Fig. $^1$H-NMR of receptor-ligand complex.** Resonances of ligand (ligand **5**) are presented.
(DOCX)

**S9 Fig. $^1$H-NMR of receptor-ligand complex.** Resonances of ligand (ligand **26**) are presented.
(DOCX)

**S10 Fig. $^1$H-NMR of receptor-ligand complex.** Resonances of ligand (ligand **27**) are presented.
(DOCX)

**S11 Fig. $^1$H-NMR of receptor-ligand complex.** Resonances of ligand (ligand **28**) are presented.
(DOCX)

**S12 Fig. $^1$H-NMR of receptor-ligand complex.** Resonances of ligand (ligand **29**) are presented.
(DOCX)

**S1 Table. Docking scores and binding affinity estimation of fragments against donor and acceptor domains of FUT2.**
(DOCX)

**S2 Table. Types of ligand-receptor interactions in donor and acceptor binding sites of receptor protein FUT2.**
(DOCX)

## Author Contributions

**Conceptualization:** Humaira Zafar, M. Iqbal Choudhary.

**Methodology:** Muhammad Atif.

**Project administration:** Humaira Zafar.

**Supervision:**  Atia-tul-Wahab, M. Iqbal Choudhary.

**Writing – original draft:** Muhammad Atif.

**Writing – review & editing:** Humaira Zafar, Atia-tul-Wahab.

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
