## [Decision Letter · Decision Letter 0]

27 Jun 2021

PONE-D-21-12616

Fucosyltransferase 2 inhibitors: Identification of new leads via docking and STD-NMR studies

PLOS ONE

Dear Dr. Zafar,

Thank you for submitting your manuscript to PLOS ONE. After careful consideration, we feel that it has merit but does not fully meet PLOS ONE’s publication criteria as it currently stands. Therefore, we invite you to submit a revised version of the manuscript that addresses the points raised during the review process.

We look forward to receiving your revised manuscript.

Kind regards,

Mohammad Shahid, Ph.D.

Academic Editor

PLOS ONE

Journal Requirements:

Reviewers' comments:

Reviewer's Responses to Questions

**Comments to the Author**

1. Is the manuscript technically sound, and do the data support the conclusions?

Reviewer #1: Yes

Reviewer #2: Yes

2. Has the statistical analysis been performed appropriately and rigorously? 

Reviewer #1: Yes

Reviewer #2: Yes

3. Have the authors made all data underlying the findings in their manuscript fully available?

Reviewer #1: Yes

Reviewer #2: Yes

4. Is the manuscript presented in an intelligible fashion and written in standard English?

Reviewer #1: Yes

Reviewer #2: Yes

5. Review Comments to the Author

Reviewer #1: Humaira Zafar et al submitted an article having the title “Fucosyltransferase 2 inhibitors: Identification of new leads via docking and STD-NMR studies” to on PLOS ONE. This article explains to identify the inhibitors for Fucosyltransferase 2 via docking and STD-NMR studies. Thus I recommended the artcle for the publication with minor revision. The following points need to be noted and revised prior to publication,

1. The authors are suggested to concise the introduction part.

2. Authors should also provide no. of interactions during docking.

3. Article needs to improve quality of the figures.

4. The authors are suggested to give the recent references in support of this work.

Reviewer #2: Response: The manuscript by H. Zafar and coworkers beautifully designed different complexes to identify the inhibitors for FUT2 and studied the cancer properties of these complexes.

The work seems to be carried out thoroughly and the compounds were characterized using various techniques. In silico-screening of natural and synthetic compounds have been performed to better understand the system. Molecular docking studies and epitope mapping in ligands further studied to better explain the interactions of ligands with amino acid residues of the active site of FUT2 and receptor protein.

The manuscript is nicely written and reads well. The introduction starts with fucosyltransferases and ends to form different complexes and their use as therapeutic agents in Cancer. Further, the drug-like properties are confirmed using swiss ADME and ToxiM analysis.I would like to suggest the Author to give the clear NMR for all the complexes, although it is already confirmed using other studies/techniques.

6. PLOS authors have the option to publish the peer review history of their article (what does this mean?). If published, this will include your full peer review and any attached files.

Reviewer #1: No

Reviewer #2: No

---

## [Author Response · Author response to Decision Letter 0]

20 Aug 2021

Reviewer 1:

Comment 1:

The authors are suggested to concise the introduction part.

Response:

The introduction part has been concised accordingly.

Comment 2:

Authors should also provide no. of interactions during docking.

Response 2:

The detail of number and types of interactions predicted during docking studies are now presented in the supplementary section Table-S3 

Comment 3:

Article needs to improve quality of the figures.

Response 3:

The quality of figures has now been improved accordingly.

Comment 4:

The authors are suggested to give the recent references in support of this work.

Response 4:

The manuscript has been updated with the recent references.

Reviewer 2:

Comment 1:

I would like to suggest the author to give the clear NMR for all the complexes.

Response:

All the clear NMR spectra for the complexes have given in the supporting information figures (S2-S12).

---

## [Editor Report · Decision Letter 1]

6 Sep 2021

Fucosyltransferase 2 inhibitors: Identification via docking and STD-NMR studies

PONE-D-21-12616R1

Dear Dr. Zafar,

We’re pleased to inform you that your manuscript has been judged scientifically suitable for publication and will be formally accepted for publication once it meets all outstanding technical requirements.

Kind regards,

Mohammad Shahid, Ph.D.

Academic Editor

PLOS ONE
---

## [Editor Report · Acceptance letter]

6 Oct 2021

PONE-D-21-12616R1 

Fucosyltransferase 2 inhibitors: Identification *via* docking and STD-NMR studies 

Dear Dr. Zafar:

I'm pleased to inform you that your manuscript has been deemed suitable for publication in PLOS ONE. Congratulations! Your manuscript is now with our production department. 

Kind regards, 

on behalf of

Dr. Mohammad Shahid 

Academic Editor

PLOS ONE